# Single-cell chromatin accessibility and lipid profiling reveals SCD1-dependent metabolic shift in adipocytes induced by bariatric surgery

Blaine Harlan[1], Hui Gyu Park[2], Roman Spektor[1], Bethany Cummings[3,4], J. Thomas Brenna[2,5], Paul D. Soloway[1,4,5]*

1 Field of Genetics, Genomics, and Development, Department of Molecular Biology and Genetics, Cornell University, Ithaca, New York, United States of America, 2 Dell Pediatric Research Institute, Department of Pediatrics, University of Texas at Austin, Austin, Texas, United States of America, 3 Department of Surgery, School of Medicine, University of California, Davis, Sacramento, California, United States of America, 4 Department of Biomedical Sciences, College of Veterinary Medicine, Cornell University, Ithaca, New York, United States of America, 5 Division of Nutritional Sciences, College of Agriculture and Life Sciences, Cornell University, Ithaca, New York, United States of America

* soloway@cornell.edu

**Data Availability Statement:** Sequencing data will be deposited in GEO at https://www.ncbi.nlm.nih.gov/geo/ (accession number: GSE191064).

## Abstract

Obesity promotes type 2 diabetes and cardiometabolic pathologies. Vertical sleeve gastrectomy (VSG) is used to treat obesity resulting in long-term weight loss and health improvements that precede weight loss; however, the mechanisms underlying the immediate benefits remain incompletely understood. Because adipose plays a crucial role in energy homeostasis and utilization, we hypothesized that VSG exerts its influences, in part, by modulating adipose functional states. We applied single-cell ATAC sequencing and lipid profiling to inguinal and epididymal adipose depots from mice that received sham surgery or VSG. We observed depot-specific cellular composition and chromatin accessibility patterns that were altered by VSG. Specifically, accessibility at *Scd1*, a fatty acid desaturase, was substantially reduced after VSG in mature adipocytes of inguinal but not epididymal depots. This was accompanied by reduced accumulation of SCD1-produced unsaturated fatty acids. Given these findings and reports that reductions in *Scd1* attenuate obesity and insulin resistance our results suggest VSG exerts its beneficial effects through an inguinal depot-specific reduction of SCD1 activity.

## Introduction

In obesity, the positive energy balance can lead to insulin resistance, chronic inflammation, cardiovascular disease, and dyslipidemia. Currently, vertical sleeve gastrectomy (VSG) is the most common bariatric surgical treatment for obesity in humans, and many effects seen in humans have been demonstrated in mouse models. VSG leads to rapid metabolic changes that precede weight loss; notably, 80% of patients with type 2 diabetes experience complete

**Funding:** The author(s) received no specific funding for this work.

**Competing interests:** The authors have declared that no competing interests exist.

remission after bariatric surgery, often within days [1, 2]. Bariatric surgery remains the most effective treatment for long-term remission of type 2 diabetes [3]. The mechanisms underlying weight loss-independent improvements are incompletely characterized. One may involve liver bile acids, which regulate glucose homeostasis through the bile acid receptor TGR5. Signaling through this pathway increases after VSG in mice [4]. Additionally, levels of GLP-1, a hormone produced by the gut, are increased after VSG, and its signaling through GLP-1R in pancreatic β-cells is important for the glucoregulatory improvements of VSG [5]. There is extensive crosstalk between the liver and adipose tissue, which affects the metabolism of glucose and lipids. However, the effects of VSG on adipose depots specifically have not been thoroughly characterized.

Adipose depots are found throughout the body and play different roles in metabolism. In humans, white adipose depots classified based on location include visceral adipose tissue (VAT) or subcutaneous adipose tissue (SAT). Excess VAT has been associated with many health risks, while excess SAT lacks these deleterious associations. Bariatric surgery has been reported to impart several changes to adipose tissue. There is a loss of fat mass, first in SAT and later in VAT, indicating the depots respond differentially to the intervention [6–8]. After VSG, levels of adiponectin rise; its elevated autocrine and endocrine activities can respectively increase fatty acid oxidation in adipocytes and inhibit liver glucose production [9]. Reduced fat mass after VSG normalizes leptin levels, which can lead to reduced food intake. Metabolic shifts in SAT and VAT adipose tissue have been reported after VSG [10, 11]. One study found increased lipolysis, lipid oxidation, and thermogenic gene expression in SAT from human patients. Some of these metabolic changes are consistent with stimulation of adipose beiging, in which white adipocytes become more similar to brown adipocytes leading to increased lipolysis, β-oxidation, and thermogenesis. Beige adipose thermogenesis, the production of heat by cellular metabolism, can increase basal energy expenditures leading to a negative energy balance, weight loss, and restoration of a healthy metabolism. While VSG alters lipid usage, the expression of UCP1, a critical factor involved in thermogenesis, appears to remain unchanged [10, 11]. However, UCP1-independent thermogenesis pathways have been described, leaving open the possibility that beiging occurs within white adipose tissue after VSG [12, 13]. Complicating these analyses are regional differences in the beiging process within the same adipose depot and between depots [14]. Overall, the mechanisms by which VSG influences adipose functions are incompletely characterized.

Because the increased energy expenditure induced by VSG may occur through non-shivering thermogenesis, we hypothesized that VSG induces changes in lipid metabolism in adipose and imparts other cell state changes in adipose tissues, particularly in mature adipocytes. Given the functional differences among adipose depots and the cell heterogeneity within depots, an understanding of the effects of VSG on adipose function requires high-resolution analyses at the cellular level.

Tissue analyses at the single-cell level can identify cell types and report changes in cellular composition and chromatin regulation resulting from VSG interventions. Most single-cell sequencing technologies rely on microfluidic devices, which are unsuitable for assaying large lipid-filled mature adipocytes, given their buoyancy in aqueous media. Chromatin accessibility in isolated nuclei can reveal, in an unbiased way, the range of cell types present in adipose depots, their abundances, and changes in their regulatory sites that arise in response to VSG. We applied this analysis to inform the mechanisms by which VSG modifies adipose and restores healthy energy homeostasis.

Our study investigated the chromatin accessibility changes in single cells of inguinal (subcutaneous) and epididymal (visceral) adipose depots after VSG or sham surgery. We identified distinct cellular compositions and gene accessibilities between these depots. Additionally, we

characterized three subtypes of mature adipocytes in both depots and identified differences in abundance and chromatin accessibility that varied with depot and surgical treatment. Altered chromatin states were observed in inguinal VSG-responsive adipocytes at genes regulating lipid metabolism. These included genes essential to β-oxidation of lipids and fatty acid saturation state. These chromatin changes were also accompanied by corresponding changes in unsaturated lipid biogenesis and a shift of white adipocytes to a thermogenic beige-like state. Previous studies have shown that these changes are protective against obesity and improve glucose metabolism, leading to the conclusion that VSG exerts its beneficial effects, in part, through modifying the metabolism of unsaturated fatty acids and increased energy expenditure through thermogenesis in a subset of adipocytes in the inguinal depot.

## Results

### snATAC libraries from inguinal and epididymal adipose tissue

To characterize the effects of VSG on adipose tissues that are independent of body mass, we used tissues from mice fed a high-fat diet for eight weeks before being subjected to VSG. As controls, we used weight-matched sham-operated animals (Fig 1A). The metabolic phenotyping for these mice has been previously reported in [15]. The white adipose depots we used for chromatin and lipid profiling studies were inguinal (subcutaneous) and epididymal (visceral), collected from three animals each.

Prior single-cell sequencing studies using white adipose tissue focused on the stromal vascular fraction (SVF), which does not contain mature adipocytes [16]. Although the buoyancy created by lipids in mature adipocytes interferes with the reliable formation of aqueous droplets used in many single-cell methods, the combinatorial indexing approach using isolated nuclei in single nuclei ATAC-seq (snATAC-seq) allowed us to study the chromatin accessibility changes in adipose tissue from mice that underwent VSG. After sequencing the snATAC libraries, removing doublets, and applying quality control cutoffs, we acquired chromatin profiles from 28,372 cells, with a minimum of 5,087 cells per group in inguinal (ING-SHAM, ING-VSG) and epididymal (EPI-SHAM, EPI-VSG) samples (S1A–S1F Fig). An advantage of ATAC-seq is the ability to examine chromatin accessibility at distal intergenic regions in addition to genes, which can identify potential regulatory elements that control cell properties and responses. We called peaks on aggregated data from each treatment group. About 25% of the peaks fell in promoter regions and 25% in distal regions (S1G Fig).

### Clustering and cell type identification of cells within adipose depots reveals distinct adipocyte subtypes

We first identified cell types by unsupervised clustering of the snATAC data after dimensional reduction. There were eleven clusters of cells in the combined ING and EPI dataset, representing 28,372 cells (Fig 1B). To classify cell types within these clusters, we used gene ontology (GO) enrichment for the top 200 (Fig 1C) or top 500 (S2C Fig) differentially accessible genes from each cluster to provide insight into biological processes. We then confirmed the cell type assignment using known marker genes (Fig 1D and 1E). Four clusters (AD1-4) collectively represented mature adipocytes, the most abundant cell type. Their top differentially accessible genes included canonical adipocyte markers such as *Pparg* and *Cebpa and* genes in GO enrichment categories that included fatty acid metabolic process.

The second largest cluster (APC) had GO enrichment categories associated with morphogenesis and WNT signaling. This cluster represented adipocyte progenitor cells (APCs) based on accessibility at known APC markers *Fbn1*, *Timp2*, *Mfap5*, *Anxa3* [17]. The next cluster

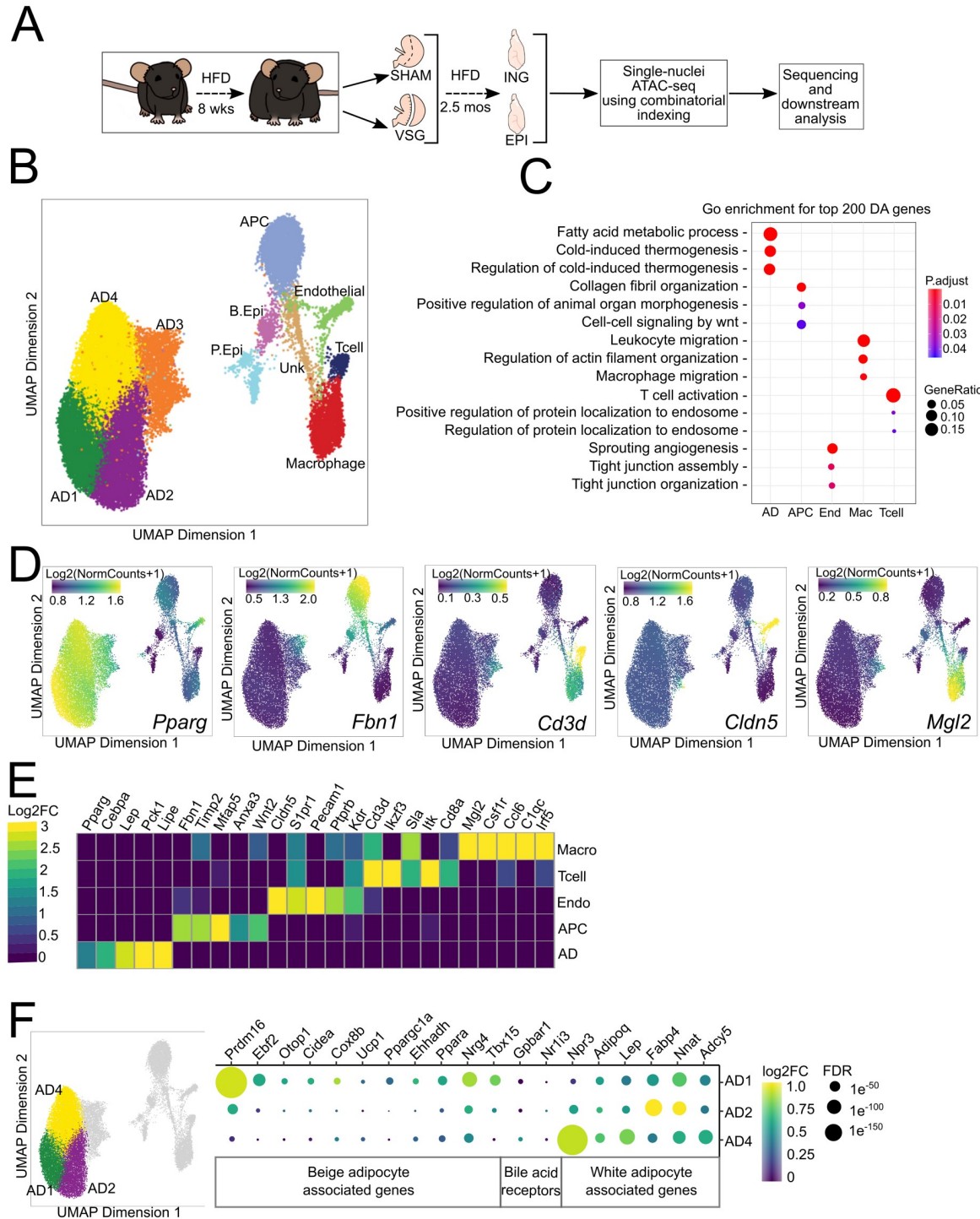

**Fig 1. Clustering and cell type identification of cells in epididymal and inguinal adipose depots.**

(End), identified as endothelial cells, had signatures of angiogenesis and tight junction enrichments and known marker genes *Cldn5*, *S1pr1*, *Pecam1*, *Ptprb*, *Kdr* [18]. Two clusters of immune cells (Mac, Tcell) had distinct gene accessibility. In the first immune cluster, we found enrichment of the GO categories T-cell activation and protein localization to the

endosome; endosomes are essential in regulating antigen receptor signaling [19]. Among the top differentially accessible T-cell markers were *Cd3d*, *Ikzf3*, *Sla*, *Itk*, *Cd8a*. The second immune cluster had an enrichment for the macrophage migration GO term and higher accessibility in *Mgl2*, *Csf1r*, *Ccl6*, *C1qc*, *Irf5*.

One unidentified cluster (Unk) did not result in any significant GO enrichments and few differentially accessible genes. This cluster also had a lower mean TSS enrichment score than other clusters, suggesting this cluster represented lower quality nuclei with high background reads. Therefore, it was excluded from further analysis. Finally, we identified two clusters (P. Epi and B.Epi) as basal epididymal cells and principal epididymal cells based on marker genes (S2B Fig) [20]. Because the epididymis is located close to the EPI depot, epididymal cells included in the EPI adipose are likely a dissection artifact, and they were not included in the analysis. While innervation in adipose tissue is vital in regulating adipose metabolism, cell body projections of neurons do not contain nuclei, and they were not present in the dataset. Overall, we found mature adipocytes, APCs, endothelial cells, T-cells, and macrophages. Since most of the cells within the dataset were mature adipocytes, this allowed us to examine regulatory changes at a single-cell level, which was not previously possible.

Among the four clusters of adipocytes, three (AD1, AD2, and AD4) had the properties of mature adipocytes, including high accessibility in canonical adipocyte markers *Pparg* and *Cebpa*. The remaining subcluster, AD3, had lower accessibility in *Pparg* than AD1,2, and 4, as well as accessibility in genes associated with non-adipocytes. Though AD3 shares some properties with the other three adipocytes, it was sufficiently distinct that we chose to exclude it from the analyses focused on mature adipocytes. AD3 is unlikely to be doublet cells as we previously removed likely doublets. AD3 may be an intermediate cell type between APCs and mature adipocytes. Because that cluster displayed no abundance or accessibility differences between depots and treatment groups, we focused our attention on AD1,2,4 in our analyses of mature adipocytes.

Recent spatially-resolved and single cell transcriptional data of human subcutaneous adipose tissue identified adipocyte subpopulations with distinct sensitivities to insulin [21]. We integrated those human transcriptomic data with our scATAC data to predict whether any of our adipocyte clusters were similar to the insulin responsive human adipocyte 2 cluster (S2D–S2F Fig). While all adipocyte clusters we identified were most like human adipocyte 1 (non-insulin responsive), AD1 had more cells predicted to be like human adipocyte 2 (insulin responsive) than either AD2 or AD4. Indicating that our adipocyte clusters correspond to similar adipocyte subtypes in humans and that AD1 may be more responsive to insulin.

To further understand the molecular basis of adipocyte heterogeneity in the inguinal depot, we compared the accessibility of genes in AD1,2,4 (Fig 1F). These clusters exhibited chromatin accessibility at marker genes for white adipose tissue, including *Npr3*, *Adipoq*, *Lep*, *Fabp4*, *Nnat*, *Adcy5*, *Psat1*, with the highest accessibility in AD2 and AD4. We also found that the beige-associated transcriptional regulators *Prdm16* and *Ebf2* were more accessible in AD1 (Fig 1F and S4A Fig). Notably, *Prdm16* is a crucial transcriptional coregulator of the beiging and the thermogenic program [22, 23]. Furthermore, *Ebf2* transcriptionally regulates brown and beige adipocyte-specific genes by recruiting chromatin remodeling complexes [24–27]. Accessibility at *Cox8b*, a regulator of mitochondrial β-oxidation, was also increased in the AD1 cluster. These data suggest that among mature adipocytes, AD1 is competent to participate in thermogenesis, as is the case for beige adipocytes. While AD1 exhibits accessibility at both white and beige marker genes, most studies show beiging leads to a loss of the white adipocyte gene expression program rather than the acquisition of a beige program. Our findings might be explained by studies showing inguinal adipocytes can interconvert between white and beige functional states depending on environmental stimuli [28]. The maintenance of open

chromatin within white marker genes in AD1 suggests that while these cells may be competent to participate in thermogenesis, they may also be poised to return to a white state in response to changing physiologic signals. Accessibility at bile acid receptor genes *Gpbar1* (TGR5) and *Nr1i3* (CAR) was not significantly different among any subclusters or treatment groups. Although TGR5 signaling is increased after VSG, this is due to increases in the bile acid ligands, which regulate glucose homeostasis through TGR5 [4]. However, beige-like AD1 cells have increased accessibility of the TGR5 target genes (*Prdm16*, *Cidea*, and *Ppargc1a)* (Fig 1F) [29].

A. Experimental design showing details of treatments and adipose tissues used for snATAC. Sham mice were pair-fed to be weight matched (WM) to VSG animals. B. UMAP plot of 28,372 cells passing quality control from inguinal and epididymal adipose tissues of VSG and SHAM mice. Colors represent 11 different clusters determined by unbiased clustering. APC, adipocyte progenitor cells; AD1, adipocyte subcluster 1; AD2, adipocyte subcluster 2; AD3, adipocyte subcluster 3; AD4, adipocyte subcluster 4; Mac, macrophage; Endo, Endothelial; Unk, unknown; B. Epi, basal epididymal epithelial; P. Epi, principal epididymal epithelial. C. Gene ontology enrichment for biological processes using the top 200 differentially accessible genes for each major cell type. D. Imputed marker gene accessibility using MAGIC overlaid on the UMAP plot. E. Validation of cell type annotation showing log2-fold change of selected cell type marker genes. FDR<0.01. F. Chromatin accessibility of beige (AD1) and white adipocyte (AD2, AD4) associated genes in the three mature adipocyte subclusters.

## Chromatin accessibility of developmental genes and cell type composition are adipose depot-specific

Having identified the mature adipocytes in our data, we determined how they differ between the ING and EPI depots. Indeed, previous studies have identified significant functional heterogeneity between white adipose depots due to their divergent developmental origins [30]. We compared cluster abundances and accessibility patterns found in cells from SHAM animals to characterize the cellular and molecular differences between the depots. Comparison of the relative abundances of mature adipocytes within each cluster revealed AD1 to be 6.5-fold more abundant in ING relative to EPI, and AD4 2.3-fold more abundant in EPI relative to ING (Fig 2A). The lack of beige-like AD1 in EPI is consistent with the reports that visceral fat depots are resistant to beiging [31]. Because there were few AD1 cells in EPI, we restricted our comparisons of mature adipocytes between ING and EPI to AD2 and AD4. Many of the differentially accessible genes in AD4 between the two depots are associated with adipocyte differentiation, such as *Zfhx4*, *Hoxa9*, *Hoxc10*, *Tbx15*, *and Mir196a/b* (Fig 2B). *Hoxc10*, which was more accessible in ING, has been shown to repress beiging in subcutaneous adipose tissue but is not expressed in brown or visceral adipose tissue [32]. Conversely, *Mir196a*, which was also more accessible in ING, was shown to increase expression of beige-associated genes through suppression of *Hoxc8*, suggesting AD4 adipocytes within the ING depot may be poised to regulate beiging [33]. Another gene, *Angptl4*, which was more accessible in ING AD2, promotes angiogenesis and allows expansion of subcutaneous depots in response to a positive energy balance [34].

In addition to finding differences in mature adipocytes within the two depots, we found differences within their precursors—the APC cluster. APCs were 1.5-fold more abundant in the EPI than the ING depot (Fig 2C). Furthermore, there was depot-specific accessibility at established regulators of adipocyte development in APCs (Fig 2D). *Wt1*, a selective marker of visceral APCs, was more accessible in EPI APCs, whereas *Tbx15* was more accessible in ING APCs [35, 36]. Notably, *Tbx15* is required for beige cell emergence in white adipose tissue and

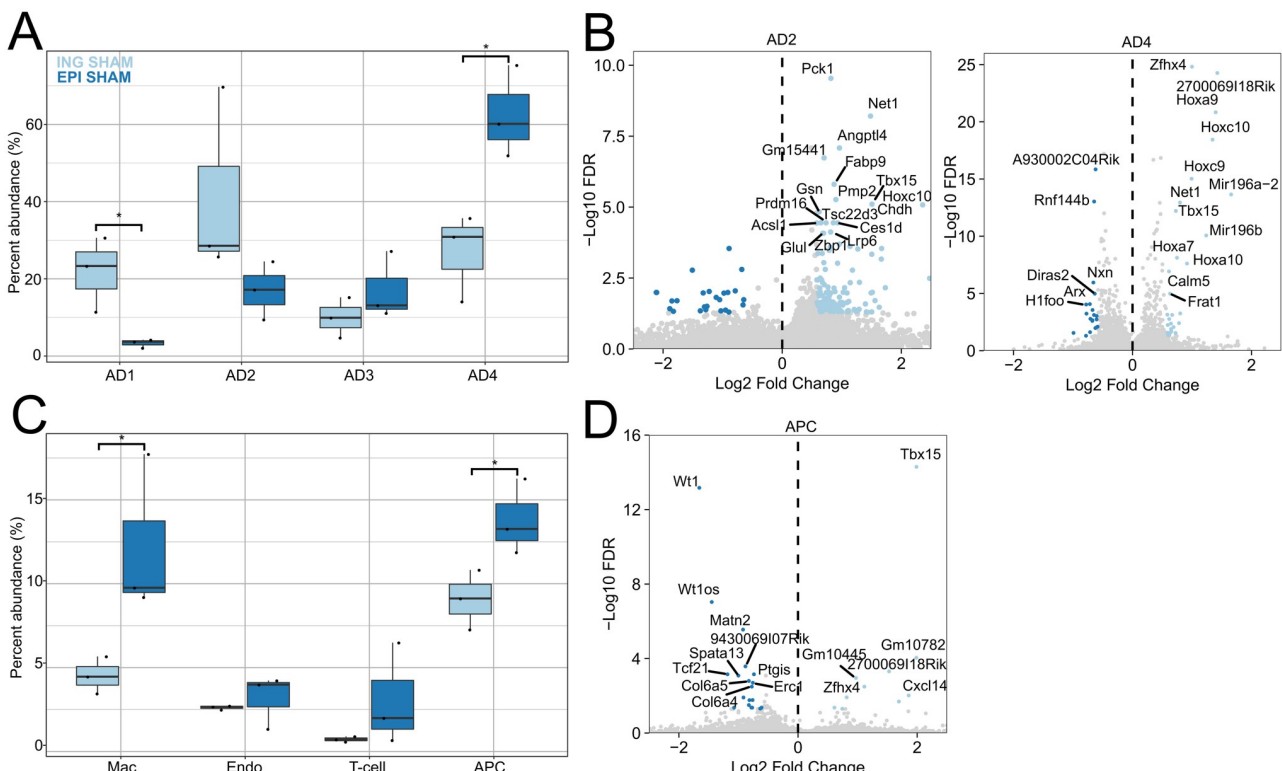

**Fig 2. Adipose depot specific differences in cells from SHAM surgery mice.** A. Adipocyte abundances in ING (light blue) and EPI (dark blue) depots of SHAM animals. Dots represent each biological replicate. (*n* = 3). B. Gene accessibility differences for AD2 (left) and AD4 (right) clusters comparing ING and EPI cells. Positive and negative log2-fold changes respectively indicate increased accessibility in ING and EPI depots. C. Abundances of endothelial, APC, T-cells and macrophages in ING (light blue) and EPI (dark blue) depots of SHAM animals (*n* = 3). D. Gene accessibility differences for APCs ING and EPI depots. Positive and negative log2-fold changes respectively indicate increased accessibility in ING and EPI depots. *p < 0.05. Two-tailed students t-test was performed for A and C.

priming of the thermogenic state [37, 38]. Collectively, the differences in accessibility of genes in APCs related to adipocyte differentiation highlight distinct mechanisms for regulating adipocyte precursors in the two depots. These same mechanisms are likely responsible for the observed difference in adipocyte abundance between ING and EPI depots and the greater propensity of cells in the ING depot to undergo beiging in response to physiologic stimuli, including VSG.

In addition to genes regulating adipocyte cell fate, we found gene accessibility changes in APCs that regulate interactions with immune cells. The adipokine *Cxcl14*, which was more accessible in ING APCs, is involved in recruiting anti-inflammatory, alternatively-activated (M2) macrophages [39]. In contrast, pro-inflammatory *Tcf21* was more accessible in EPI APCs. These findings demonstrate limited inflammatory signals in the ING depot relative to the EPI depot. The macrophage cluster was more abundant in EPI adipose tissue; however, no genes were differentially accessible. There were no differences in abundance or accessibility of genes in T-cells or endothelial cells between the depots.

## Changes in accessibility at transcription factor motifs identify regulatory changes in inguinal VSG adipocytes

We next analyzed the effects of VSG on both cell abundances and transcription factor (TF) motif enrichments at sites of differential accessibility. The latter focus was motivated by the

goal of understanding the regulatory mechanisms underlying VSG responses. Overall, the distribution of cell abundances in clusters did not change significantly between VSG and SHAM in either depot (S4B Fig). While adipocyte progenitor cells and macrophages are known to change in response to obesity, we observed few cell number or accessibility changes between VSG and SHAM cells in ING and EPI adipose tissue for these cell types or T-cells. This may indicate that immune cells and APCs are not involved in physiologic changes associated with VSG. However, these cells represented fewer than 15% of the total cells analyzed from the four groups of mice, and accordingly, subtle but relevant changes may not be revealed in our data.

To identify potential gene regulatory changes that accompany VSG, we first characterized the transcription factor motifs enriched in adipocytes from the ING and EPI depots at regions of differentially accessible chromatin. For these analyses, we utilized the combined data from all classes of adipocytes, as limiting the analyses to individual subsets of adipocytes reduced our overall power. We first called peaks on the combined dataset, then determined differentially accessible peaks in VSG and SHAM-ING adipocytes. We found 13,449 peaks with increased accessibility and 4,943 with decreased accessibility in VSG mice (Fig 3A). In peaks with increased accessibility, there were enrichments for STAT5A/B, NFIC/X, and STAT1,3,4 motifs (Fig 3B and 3C). Importantly, adipocytes express STAT1,3,5A, 5B, and 6; therefore, the differential accessibility at motifs for those factors is likely to have functional consequences [40]. Notably, STAT5A directly binds to the promoter of *Acox1*, the first enzyme of the fatty acid b-oxidation pathway, which catalyzes the desaturation of acyl-CoAs to 2-*trans*-enoyl-CoAs [41, 42]. This suggests that among the effects of VSG is elevated *Stat5*-dependent fatty acid b-oxidation. In peaks with decreased accessibility in adipocytes from VSG mice, there was an enrichment of motifs for the progesterone receptor (PGR) and the glucocorticoid receptor (NR1C3) (Fig 3B). Consistent with this, in VSG mice, we found reduced accessibility within genes known to be regulated by glucocorticoids such as *Serpine1*, *Fkbp5*, *Scd1* (Fig 4A). Because glucocorticoids are associated with stress responses and promote obesity and insulin resistance, this further suggests that additional effects of VSG include reduction in glucocorticoid-mediated stress responses and altered regulation of glucocorticoid target genes that influence metabolism.

Transcription factor motifs enriched at differentially accessible peaks indicate their potential for regulating nearby target genes. To determine which genes could be regulated by the top enriched motifs, we located genes within 100kb of differentially accessible peaks containing STAT5A, NFIC, and NR1C3 motifs and then identified the biological processes they control as indicated by GO enrichment (Fig 3D). Surprisingly, each TF motif resulted in different GO enrichment categories for VSG and SHAM, indicating that coregulatory factors may modulate transcription factor binding under the physiological conditions affected by VSG, and that the coregulators may shift the effects of the TFs. In adipocytes from mice subjected to VSG, GO enrichments at putative target genes near sites of increased STAT5A, NFIC, and NR1C3 motif accessibility were associated with adaptive thermogenesis, cold-induced thermogenesis, and fatty acid oxidation. Notably, we observed several peaks with STAT5A motifs near *Elovl5* (S3B Fig), a fatty acid elongase. In contrast, genes near peaks with significantly decreased accessibility were associated with inflammation (inflammatory response, interleukin-1 mediated signaling pathway, stress-activated MAPK cascade) GO enrichment categories. The genes associated with STAT5A SHAM peaks did not produce significant GO enrichments. These findings are consistent with the facts that improvements in metabolism and reductions in inflammation are associated with VSG, and they strongly implicate the functional importance of these specific TFs in adipose homeostasis impacted by VSG.

TF motif enrichment identifies potential TF binding regions, not all of which are bound. Therefore, we sought to validate whether regions with NR1C3 motifs were indeed bound by

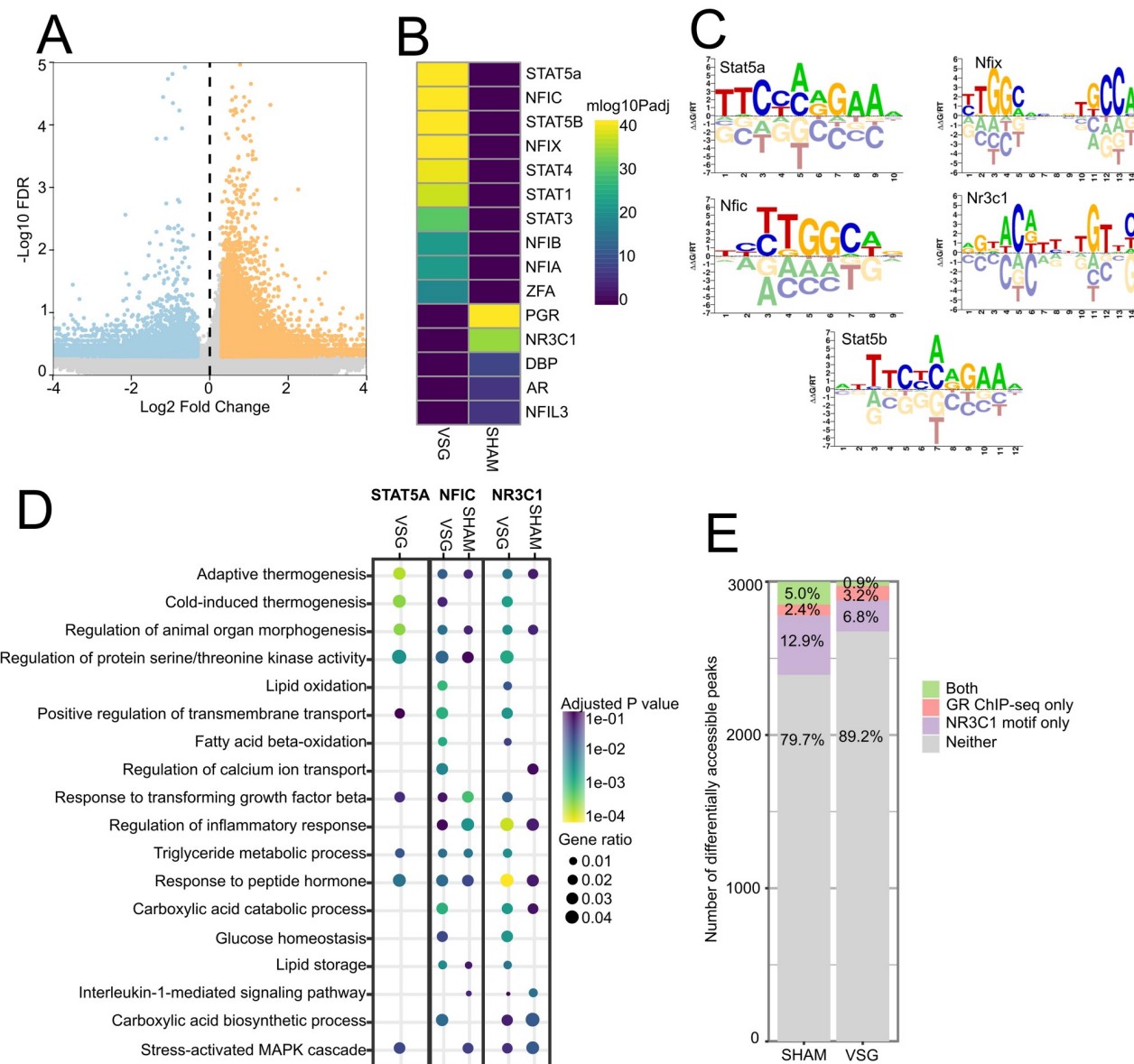

**Fig 3. Differential accessibility of peaks with transcription factor motifs in response to VSG.** A. Volcano plot of differentially accessible peaks comparing VSG to SHAM ING adipocytes. The 13,449 peaks with positive fold change indicate increased accessibility in VSG; the 4,943 peaks with negative fold change indicate decreased accessibility in VSG. FDR < 0.5 and absolute Log2-Fold Change> 0.3. B. Transcription factor motif enrichment in differentially accessible peaks using the CISBP motif database. VSG (peaks with positive fold change) and SHAM (peaks with negative fold change). C. Motif sequence logos for the top enriched motifs. D. Gene ontology enrichment for biological process using the nearest genes (<100kb) to peaks with top motifs. Adjusted P.value < 0.1 for GO enrichments. E. Overlap of the top 3,000 differentially accessible peaks for VSG and SHAM ING adipocytes containing NR3C1 motifs, and glucocorticoid receptor ChIP-seq binding regions reported in Yu et al., 2010.

the glucocorticoid receptor (GR) in adipocytes and if there were differences between the treatment groups. Yu *et al.* identified GR binding regions in 3T3-L1 differentiated adipocytes using ChIP-seq [43]. They found that many of the GR binding sites were near genes involved in triglyceride homeostasis. Several of these were differentially accessible in our dataset, including *Scd1* (Fig 4A). To determine whether the motifs we found at open chromatin in adipose overlap with known GR binding regions in 3T3-L1 adipocytes, we first identified the subset of the

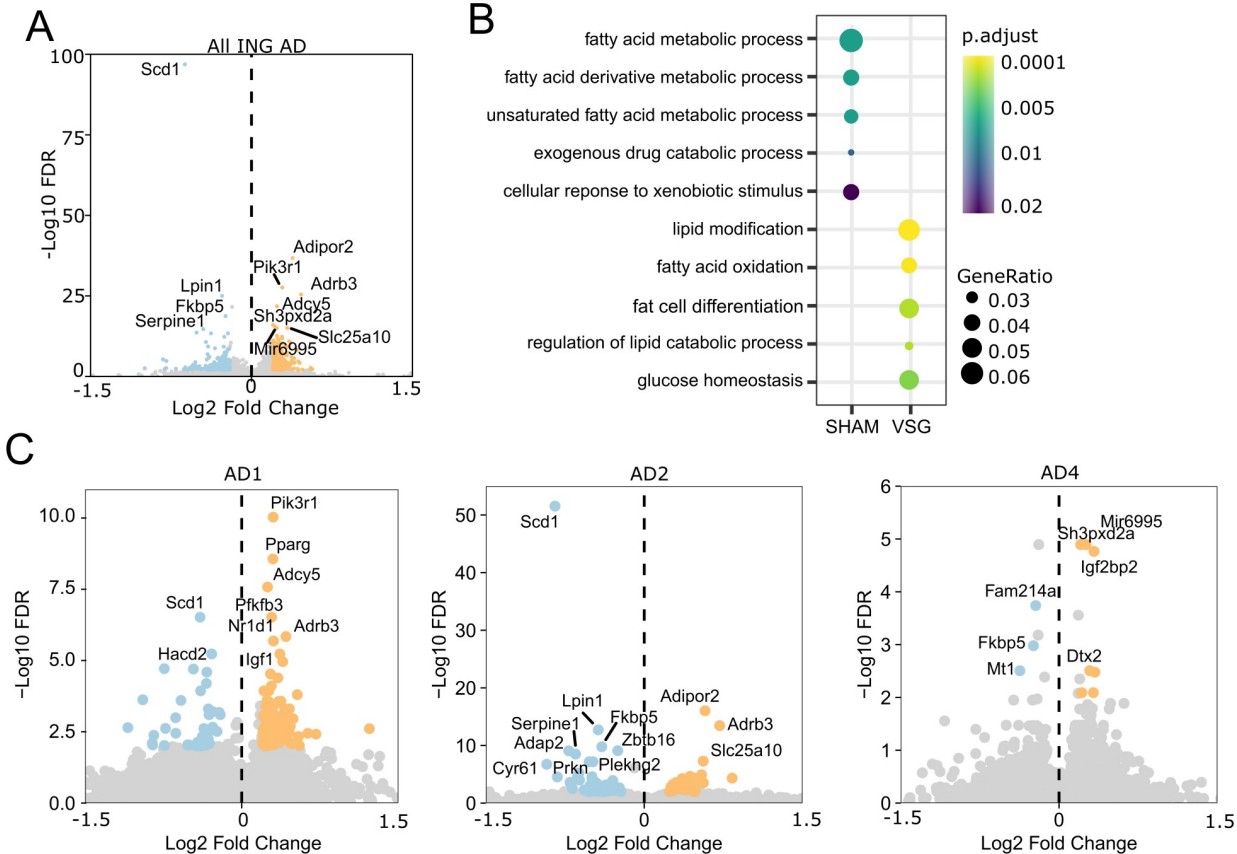

**Fig 4. Accessibility changes in genes after VSG in inguinal mature adipocytes.** A. Volcano plot of differentially accessible genes comparing VSG to SHAM ING adipocytes. Positive fold change indicates increased accessibility in VSG cells. B. Biological process gene ontology enrichment for the top 200 most significantly differentially accessible genes. C. Volcano plots of differentially accessible genes in VSG vs SHAM in each mature adipocyte cluster.

top three thousand peaks in ING adipose that were both differentially accessible in VSG *vs.* SHAM mice and that had NR1C3 motifs, and then we intersected them with the GR ChIP-seq data from 3T3-L1 adipocytes. We found that 5.8% of peaks with NR1C3 motif enrichment in SHAM adipose overlapped with GR binding regions in 3T3-L1 adipocytes, whereas only 0.8% of the VSG peaks overlapped (Fig 3E). This difference in overlap suggests that accessible NR1C3 motifs in SHAM adipose are likely to include more functional GR binding sites than VSG adipose. Additionally, this difference may in fact understate the importance of GR binding sites in SHAM adipose, as the top peaks in SHAM adipose had more NR3C1 motifs (4.5% ≥ 2 motifs) than did VSG adipose (1% ≥ 2 motifs). While *in vitro* cultured 3T3-L1 adipocytes do not replicate cell states found in adipose tissue, these findings are consistent with our interpretation that decreased gene binding and activation by GR is associated with adipocyte responses to VSG *in vivo*, linking this change in GR regulation with the beneficial effects of VSG.

## VSG induces gene accessibility changes in ING mature adipocytes regulating thermogenesis, insulin sensitivity, and lipid metabolism

Many of the transcription factor motifs that were differentially accessible at intergenic sites of mature adipocytes from VSG mice, including those associated with GR binding, were found

near genes essential to adipocyte metabolism. Therefore, we extended our analyses to consider additional functions of the genes nearest the differentially accessible peaks and to consider intragenic differential chromatin accessibility in adipocytes. Among the most differentially accessible genes in VSG *vs*. SHAM adipocytes were *Adipor2*, *Pik3r1*, *Adrb3*, *Adcy5*, which were significantly more accessible in VSG adipocytes, and *Fkbp5*, *Serpine1*, *Scd1*, *Lpin1*, which were significantly more accessible in SHAM adipocytes (Fig 4A). Each of these genes has been implicated in the metabolic regulation of adipose. Their differential accessibility identifies pathways such as fatty acid and lipid metabolism that are influenced by VSG and likely mediate its effects (Fig 4B).

After bariatric surgery, the well-studied adipokine, adiponectin, is increased in the plasma, and its receptor, *Adipor2*, is also upregulated in SAT [44]. Our finding that VSG increases *Adipor2* accessibility is consistent with these results and supports an autocrine role of adiponectin in improved adipose insulin sensitivity [44–46]. *Pik3r1*, encoding a regulatory subunit of PI3K, is required for insulin signaling transduction, and accordingly, important for insulin sensitivity. Increased accessibility in these two insulin-sensitizing genes suggests a role for the ING depot in the improved insulin responsiveness after VSG. The sympathetic nervous system activates the b-3 adrenergic receptor, ADRB3, promoting beiging within white adipose tissue. ADRB3, but not ADRB1, stimulates a shift of mature white adipocytes into a beige state instead of *de novo* adipogenesis of beige adipocytes [47]. The increased accessibility at *Adrb3* in adipocytes from VSG mice implicates its role, and beiging, in VSG responses.

Genes with reduced accessibility in VSG are associated with metabolic dysfunction and insulin resistance. *Fkbp5* is highly upregulated by glucocorticoid stress-response hormones in adipose tissue and is implicated as a critical gene in glucocorticoid-induced insulin resistance. Its increased accessibility in adipocytes from SHAM-treated mice is consistent with the genome-wide increase in accessibility of GR binding sites in the genome of those animals. *FKBP5* is also upregulated in humans with type 2 diabetes. Notably, a polymorphism in *FKBP5* in humans is associated with variations in weight loss outcomes after bariatric surgery [48]. While the proteinase inhibitor *Serpine1* (also called PAI-1) is predominantly expressed in visceral depots, obesity increases the subcutaneous expression of *Serpine1*. High levels of plasma PAI-1 are predictive of insulin resistance, type 2 diabetes, and hepatic steatosis. That *Serpine1* accessibility is higher in SHAM adipocytes is also consistent with its being a VSG target that confers benefits. In opposition to the other genes with reduced accessibility in VSG, *Lpin1* is not associated with metabolic dysfunction or insulin resistance. Instead, it is involved in the storage of lipids as triglycerides.

The most significantly differentially accessible gene in our dataset, with reduced accessibility in VSG adipocytes, was *Scd1*, a desaturase that converts saturated fatty acids (SFAs) into monounsaturated fatty acids (MUFAs). Notably, deletion of the *Scd1* gene in mice leads to a wide array of changes in adiposity, energy storage and metabolism, and glucose homeostasis, each of which is highly relevant to responses to VSG [49–55]. Overexpression of this gene is associated with obesity in humans in addition to its effect on insulin sensitivity [56]. The functional importance of this accessibility change following VSG is described further below. Overall, the observed differentially accessible genes in VSG indicate a shift towards increased insulin sensitivity and thermogenic potential, both of which are among the benefits of VSG. Accordingly, our findings reveal molecular mechanisms underlying these benefits.

Because our data identified distinct mature adipocyte types, we wondered if the gene accessibility patterns described above for VSG and SHAM animals were present in specific adipocyte types. We found that beige-like AD1 harbored the observed accessibility changes for *Pik3r1*, *Pparg*, and *Adcy5*, while AD2 harbored the accessibility changes for *Scd1*, *Adipor2*, *Adrb3*, whereas there were fewer differences in gene accessibility of AD4 (Fig 4C). This finding

suggests that the AD4 subtype has less plasticity in responses to metabolic changes. Surprisingly, the white-like AD2, as opposed to beige-like AD1, had the most significant increase in beige-associated *Adrb3* accessibility in VSG mice. This suggests that regardless of the treatment group, beige-like AD1 cells can respond to β3-adrenergic receptor activation; however, white-like AD2 cells can acquire it after VSG. AD1 and AD2 also exhibited the most significant change in *Scd1* of the subclusters.

In summary, ING VSG adipocytes have increased accessibility at genes associated with thermogenesis and beiging. VSG also leads to decreased accessibility in genes related to insulin resistance and type 2 diabetes, such as *Scd1*, *Serpine1*, and *Fkbp5*, and these changes are found in specific adipocyte subtypes.

## Reduced SCD-1 accessibility decreases the accumulation of unsaturated fatty acids in the ING depot

Our data identified genes enriched with open chromatin for each of the four types of samples analyzed, including both depots and both treatment groups (S4D Fig). However, the most significant change in our dataset is *Scd1*, which is more accessible in AD2 adipocytes in ING SHAM (Fig 4C). SCD1 converts saturated fatty acids, stearate and palmitate, into monounsaturated fatty acids, oleate and palmitoleate, respectively. *Scd1* inhibition leads to an accumulation of stearate but not palmitate; this is partly due to the upregulation of fatty acid elongase 6 (*Elovl6*), which converts palmitate to stearate [57]. To validate whether these gene accessibility changes in ING after VSG have functional consequences on fatty acid profiles, we used mass spectrometry after one-step lipid extraction and methylation on the same adipose tissues used for our snATAC libraries (Fig 5A). This method characterized the fatty acid chain length and position of double bonds and their quantitative levels (Fig 5B). If reductions in *Scd1* accessibility in ING VSG mice have functional consequences, we predicted there to be elevated levels of SFAs or reduced levels of MUFAs, leading to a lower overall ratio of MUFAs to SFAs in the ING depot of VSG mice. We further predicted that in EPI samples, which showed no accessibility changes at *Scd1*, there would be no significant alterations in the levels or ratios of lipid species between the treatment groups. These are precisely what we observed, demonstrating the functional consequences of *Scd1* accessibility changes in the ING depot arising as a result of VSG (Fig 5C–5F). The desaturation index, which reports the ratio of MUFA to SFA for a fatty acid of a given chain length, was increased in the VSG ING depot by the accumulation of 16:0, and a reduction in 18:1n-9 lipids. However, these effects may be driven by the higher abundances of 16:1n-7 and 18:0, relative to the saturated and unsaturated forms of 16- and 18-carbon fatty acids, respectively. Importantly, in humans, the reductions in the desaturation index of 18-carbon fatty acids are positively correlated with insulin resistance [56]. These results indicate that the regulation of *Scd1* plays a vital role in the metabolic benefits of VSG through alterations in fatty acid modifications leading to improved insulin sensitivity (Fig 5G).

## Discussion

We applied snATAC-seq profiling to mouse adipose tissues from weight-matched mice fed a high-fat diet for eight weeks before VSG or sham surgery, followed by tissue collection ten weeks after surgery (Fig 1A). Using this approach, we characterized the cellular composition of adipose, uncovered differences between visceral and subcutaneous depots, revealed functionally relevant effects of VSG upon each of these variables, including impacts on fatty acid metabolism, and identified regulatory motifs that are associated with these biological differences. These findings revealed developmentally- and environmentally influenced functional physiologic responses of adipose tissues. Because VSG imparts lasting changes that attenuate

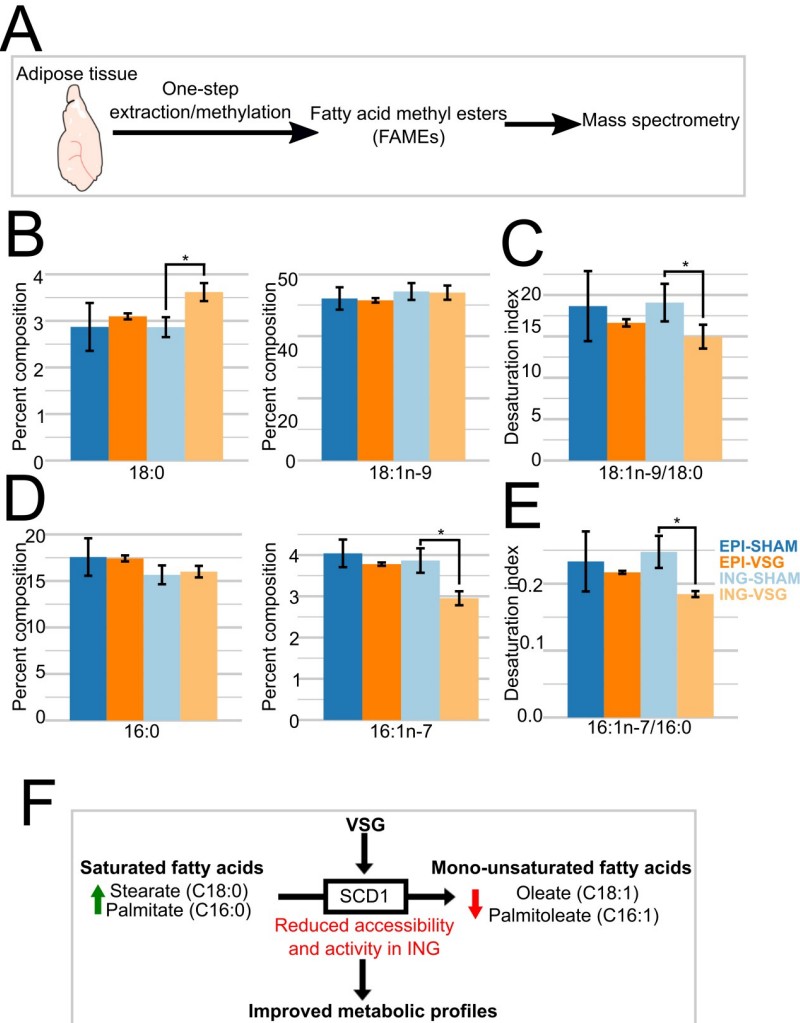

**Fig 5. Reduced accessibility in Scd1 leads to less accumulation of unsaturated fatty acids.** A. Experimental design of determining FAME profiles, using three samples per depot and treatment group. B. 18-carbon fatty acid profiles, and C, corresponding desaturation index. ($n = 3$). D. 16-carbon fatty acid profiles, and E, corresponding desaturation index. ($n = 3$). *$p < 0.05$. Two-tailed students t-test was performed for (C,D,E,F) comparing VSG to SHAM within ING or EPI adipose samples. F. Model for the contributions of VSG to metabolic health through functional accessibility changes at *Scd1*.

pathologies associated with obesity and diabetes, these findings also provide insights into functional pathways relevant to metabolic health, disease states, and the mechanisms underlying the benefits of bariatric surgical treatments for obesity.

Several challenges arise in the application of single-cell methods to whole mature adipose tissues. Mature adipocytes are large, lipid-filled, and buoyant in aqueous solutions, making droplet-based approaches difficult for their analyses. While the stromal vascular fraction of adipose does not have this issue, it includes only a portion of cells present in adipose and cannot reveal the depot- and VSG-specific changes in mature adipocytes we report. Nuclei extraction methods that worked well in our hands for many tissues from mice and humans led to clumping and poor yields with adipose. Optimizations described in the methods section circumvented these issues.

Our analyses identified four classes of mature adipocytes (AD1, 2, 3, 4), which collectively were the most abundant category of cells in adipose; we also identified adipocyte progenitors, endothelial cells, T-cells, and macrophages (Fig 2A). In sham-operated mice, EPI and ING depots exhibited significant differences in the distributions of the mature adipocytes. AD1 was significantly more abundant (more than a 6-fold increase) in ING and had elevated accessibility relative to other adipocyte classes at *Prdm16*, a transcriptional cofactor that upregulates thermogenic genes. In contrast, AD4 was significantly more abundant in EPI (2-fold increase) and had elevated accessibility relative to other adipocyte classes at *Npr3*, a negative regulator of thermogenic genes (Fig 3A, [58]). These findings indicate that AD1 are thermogenic beige-like adipocytes, AD4 are white adipocytes, and that these cellular differences, in part, account for the higher beiging potential in ING *vs.* EPI depots. We found further evidence for the higher beiging potential of ING depots: Within the AD4 adipocytes, those from ING depots had higher accessibility at *Mir196a*, which positively regulates beiging through suppression of *Hox* family genes. Interestingly, *Hoxc10*, a target of *Mir196a* and a negative regulator of beiging, is only expressed in subcutaneous adipose, like ING depots, but not in visceral, like EPI, or brown adipose. Additionally, *Hoxc10* had higher accessibility in the AD4 population in ING relative to EPI. This indicated that AD4 adipocytes in ING depots have plasticity absent in EPI depots, enabling AD4 in ING to control the extent of beiging through positive and negative regulation. Furthermore, signaling triggers, such as those provided by VSG [32, 33], can direct this plasticity in ING AD4 cells, leading to beiging.

In addition to these depot-specific differences in adipocyte abundances and chromatin states, the next most abundant cell type, APCs, were present in larger numbers in EPI than in ING. Genes involved in the regulation of cell fate, *Wt1*, and *Tbx15*, were also differentially accessible in APCs from the two depots, consistent with reports of the disparate developmental origins of these two depots [35, 36]. These depot-specific APC differences may account for the differences seen in mature adipocytes derived from them in the two depots. Specifically, the increased accessibility in ING APCs at *Tbx15*, which is required for the beiging of white adipose tissue and increased thermogenesis [37, 38], may enable APC-derived mature adipocytes in the ING depot to undergo beiging in response to VSG. In addition, regulators of macrophage state were differentially accessible in APCs from the two depots. EPI APCs showed higher accessibility at *Tcf21*, which has been linked to elevated IL-6 production by APCs in visceral adipose, which can induce M1-type pro-inflammatory macrophages [59]. In contrast, ING APCs showed higher accessibility at *Cxcl14*, which has been implicated in beiging, thermogenesis and induces M2-type anti-inflammatory macrophages [39]. These differences in APCs between the two depots can, in part, account for the health hazards associated with accumulations of visceral as opposed to subcutaneous adipose.

There were significantly more macrophages and APCs in EPI than in ING, but we detected no abundance differences for endothelial cells or T-cells or depot-specific accessibility differences for these cell types. Failure to detect accessibility differences may be due to their lower cell abundances relative to adipocytes and APCs.

Subcutaneous and visceral adipose depots are known to respond differently to bariatric surgery in humans. Reductions in adiposity occur first in subcutaneous, followed by visceral depots [6–8]. Consistent with human clinical findings, VSG induced different responses in ING *vs.* EPI depots in mice. We found many accessibility changes in ING in response to VSG two and a half months after surgery and fewer effects in EPI; most changes affected adipocytes. The most significant and highest magnitude changes to chromatin accessibility were in ING AD1 and AD2 adipocytes, with VSG increasing accessibility at genes affecting insulin sensitivity, lipid metabolism, and thermogenesis. In EPI adipocytes, VSG also increased accessibility at *Pfkfb3*, which increases insulin sensitivity. These findings identify changes in gene

regulation that can contribute to the health benefits associated with VSG, including improvements in energy expenditure and glucose control. There were no significant changes within the APC population in response to VSG.

*Scd1* encodes a fatty acid desaturase that catalyzes the rate-limiting step in the production of MUFAs from SFAs; it was the gene most significantly affected by VSG, showing reduced accessibility in AD1 and AD2 adipocytes after VSG. Accompanying this reduced *Scd1* accessibility were reductions in the accumulation of MUFAs, or increased accumulation of SFAs in inguinal adipose from VSG mice, leading to reductions in the desaturation index of lipids present in the ING depot. Independent lines of evidence from *in vivo* and *in vitro* studies demonstrate that reduced SCD1 activity, and SFAs improve glucose and energy metabolism, adiposity, insulin responses, and adipose inflammation associated with obesity. Each of these is consistent with our findings implicating the importance of reduced *Scd1* accessibility in mediating the beneficial effects of VSG. Genetic studies in mice have validated the metabolic benefits of reduced *Scd1* expression *in vivo* for obesity- and diabetes-related traits. Mice bearing *Scd1*-inactivating mutations have lower fat mass, increased insulin sensitivity and signaling, lower white adipose inflammation, and higher metabolic rates than their wildtype counterparts [53, 55, 60, 61]. These effects are more dramatic in *ob/ob* leptin-deficient animals, where *Scd1* deficiency also reduces body mass and liver triglycerides [51], and improves insulin sensitivity [52]. SCD1 loss has also been associated with increased thermogenesis [50], fatty acid β-oxidation in muscles [49], and protection from carbohydrate-induced obesity [54]. These effects are likely due to the reduced fatty acid desaturation index that accompanies *Scd1*-inactivation, and not due to some secondary effect of SCD1 that is unrelated to its production of MUFAs. *In vivo* evidence comes from studies demonstrating that in *Scd1*-deficient mice, dietary oleate partially compensated for the hypothermia and elevated plasma glucose and liver glycogen stores [50].

Our findings identified cell- and gene-focused mechanisms that distinguish adipocyte responses to VSG. By extending our analyses to accessibility changes at intergenic regions, we were also able to identify gene regulatory mechanisms that may be responsible for these differences. STAT family transcription factor binding motifs were significantly enriched at regions of increased accessibility in adipocytes from VSG-treated mice. These factors are expressed in adipocytes, and in the case of STAT5a, regulates genes also found to exhibit increased accessibility in VSG, including those involved in fatty acid elongation, β-oxidation, and thermogenesis. In VSG mice, there were regions with reduced STAT5a accessibility, and these were proximal to genes with pro-inflammatory processes. These findings identify STAT5a as central to the beneficial effects of VSG and highlight that its binding is likely subject to regulatory cofactors that are important components of VSG responsiveness. We also identified motifs enriched at sites with reduced accessibility in adipocytes after VSG. These include the glucocorticoid receptor, NR1C3, which is associated with stress responses, impaired glucose metabolism, and obesity. Inferences from 3T3-L1 ChIP data indicate that the reduced accessibility at these motifs is associated with reduced occupancy by NR1C3 [43]. Importantly, glucocorticoids participate in the regulation of *Scd1*. Accordingly, the reduced accessibility of NR1C3 motifs after VSG is likely to contribute to the beneficial changes in the fatty acid metabolism we observed. Collectively, these motif findings identify likely regulatory mechanisms that are central to the effects of VSG.

## Methods

### Mice

At 2 months of age, C57/BL/6J mice were placed on a 60% energy from fat high-fat diet (HFD) for eight weeks then underwent VSG, or sham surgery as described in McGavigan et al. The

metabolic phenotyping for these mice has been previously reported [15]. The present study used archived materials from the earlier one, requiring no further institutional animal use approvals. Mice were maintained on HFD throughout the study, euthanized at 2.5 months after surgery, and the inguinal (ING) and epididymal (EPI) adipose depots were dissected without removing lymph nodes, frozen, and stored.

## Nuclei isolation

Whole adipose depots were finely minced on dry ice. Then 80mg of adipose tissue was transferred to a pre-chilled Dounce homogenizer (Wheaton #357546) with 30 mL of HB (320 mM sucrose, 0.1 mM EDTA, 0.1% NP40, 5 mM $CaCl_2$, 3 mM $Mg(Ac)_2$, 10mM Tris pH7.4, protease inhibitors (Pierce #88666), 0.016 mM PMSF). Samples were processed in groups of two (one SHAM and one VSG) to ensure any variation due to library preparation is equally applied to both SHAM and VSG. All steps were performed on ice or at 4˚C. The tissue was first homogenized with 10–15 strokes using a loose pestle then an additional 10 strokes with the tight pestle. The homogenized tissue was then filtered through a 70 μm cell strainer (BD Biosciences #352350) and centrifuged for 5 mins at 500g in a centrifuge with a swinging bucket rotor. The supernatant was removed, the cell pellet washed with ATAC-RSB (10 mM Tris-HCl pH 7.4, 10 mM NaCl, 3 mM $MgCl_2$, $H_2O$), then centrifuged for 10 minutes at 750g. After removing the supernatant, nuclei were resuspended in 1 mL of OMNI-ATAC buffer (10 mM Tris pH7.4, 5 mM $MgCl_2$, 10% DMF, 33% 1x PBS(no Ca++ or Mg++), 0.1% Tween-20, 0.01% Digitonin (ThermoFisher #BN2006)), counted on a hemacytometer, and the concentration was adjusted to 100,000 nuclei/mL.

## Library preparation

The snATAC library preparation protocol was based on Preissl et al. and Spektor et al. with some modifications [62, 63]. For tagmentation with Tn5 transposase, 8 μL of the nuclei suspension was added to each well of a 96 well plate. Then, 1 μL of ME-C barcoded transposome loaded Tn5 and 1 μL of ME-D barcoded transposome loaded Tn5 were added to each well. Tagmentation was carried out at 50˚C for 30mins. 10 μL of 40 μM EDTA was added to each well, gently vortexed, and incubated for 15 minutes at 37˚C to inactivate Tn5 transposase. 20μL of sorting buffer (2% BSA, 2mM EDTA in PBS (no Ca++ or Mg++)) was added to each well, and all wells were combined. For the freeze-thaw test, nuclei were centrifuged for 5 mins at 500g and resuspended in 500 μL of freeze buffer (10% DMSO, 90% sort buffer) and slowly frozen and stored at -80˚C.

Nuclei were stained with Draq7(1:100; Abcam #ab109202), filtered through a 30 μm filcon cell strainer (BD Biosciences #340597), and 25 nuclei were sorted into each well of a 96 well plate containing 16.5 μL of sort EB (10mM Tris pH 8, 2% BSA, $H_2O$) per well using a BD FACSAria Fusion. For PCR amplification, 2 μL of 0.2% SDS was added to each well and incubated for 7 minutes at 55˚C. Next, 2.5 μL of 10% Triton X-100, 2 μL of 25 μM Primer i5, 2 μL of 25 μM Primer i7, and 25 μL of 2X PCR mix with Q5 DNA polymerase (NEB #M0491S) (Q5 5X buffer, dNTP, Q5 enzyme, GC enhancer, $H_2O$) was added to each well. All PCR reactions were run in the same thermocycler with the following program: 72˚C 5 minutes, 98˚C 30 seconds, 15 cycles of (98˚C 10 seconds, 63˚C 30 seconds, 72˚C 1 minute), and 72˚C 5 minutes. After PCR, all wells were pooled, then 24 mL buffer PB (5:1, Qiagen #19066) and 1.2mL NaOAc were added sequentially. Samples were each run through a separate MinElute column (Qiagen #28004) using a QIAvac apparatus (Qiagen #19413), then washed twice with 750 mL Buffer PE (Qiagen #19065). The sample was then eluted in 40 uL of Buffer EB (10 mM Tris pH 8) and centrifuged at max speed on a tabletop centrifuge for 2 minutes. The following size

selection steps were performed at room temperature. First, large fragments were removed with 0.5X vol AMPure XP (Beckman Coulter #A63880) purification, and then size selected 3 times with 1.5X vol AMPure XP consecutively to remove PCR primers according to the manufacturer's protocol.

## Library quality control and sequencing

Each library was measured using a bioanalyzer for the expected nucleosomal patterning. Libraries were pooled using digital PCR on a Bio-Rad QX200 droplet digital PCR system using the following oligos: forward, AGCAGAAGACGGCATACGAGAT; reverse ATACGGCGACCACC GAGATC; internal /56-FAM/TCTTATACACATCTGAGGCGG/3BHQ_1/ [64]. Then, sequenced on a NextSeq500 mid lane output PE 150bp. A custom sequencing recipe was run; read 1 [36 imaged cycles], index 1 [8 imaged cycles, 27 dark cycles, 8 imaged cycles], index 2 [8 imaged cycles, 21 dark cycles, 8 imaged cycles], read 2 [36 imaged cycles]. Data are available through https://www.ncbi.nlm.nih.gov/gds/ under the following accession numbers: xxxx.

## Data pre-processing

Libraries were preprocessed and aligned as in Preissl et al. Reads were aligned to the mm10 genome using Bowtie2 in paired-end mode (-p 5 -t -X2000—no-mixed—no-discordant–mm) then duplicated reads and reads with MAPQ<30 were removed. The doublet score and TSS enrichment score were then calculated using ArchR [65]. After removing doublets, barcodes with fewer than 1000 fragments and a TSS score of less than 4 were removed.

## Computational analysis of snATAC-seq data

The ArchR analysis pipeline was used for most of the data analysis. After doublet removal and quality control cutoffs, dimensionality reduction was performed using ArchR's iterative LSI-SVD method. After dimensionality reduction, we did not observe significant variation between our biological replicates so additional batch correction, such as Harmony, was not performed.

A 10kb tile matrix was used for clustering and UMAP visualization. For cell type identification, ArchR's getMarkerFeatures() was used to find differentially accessible genes for each cluster. The gene score count matrix was used for all gene accessibility analyses. GO enrichment was performed by ClusterProfiler's compareCluster with a p-value cutoff of 0.05. MAGIC was used for gene accessibility imputation for gene score visualization of specific marker genes. Peaks were identified using default parameters in ArchR, which call peaks using MACS2 on each cluster, then all peaks were merged using ArchR's iterative merging approach. For motif enrichment, ArchR's hypergeometric enrichment test peakAnnoEnrichment() was used with motif annotations from the CIS-BP motif database.

## GR motifs and GR ChIP-seq comparison

Data from Yu et al. 2010 was downloaded from GEO (GSE24105). The ChIP-seq peaks were then lifted over from mm9 to mm10 annotation. Any overlap with the top 3,000 peaks from ING-VSG and ING-SHAM were counted.

## Spatial transcriptomics integration

The baseline spatial transcriptomics data were downloaded as a Seurat object and subsetted to include only adipocyte 1, 2 and 3 clusters as annotated by Bäckdahl et al [21]. Mouse gene names were converted to human using the biomaRt package. Next, the ArchR object

containing the scATAC data and the Seurat object were integrated using the addGeneIntegrationMatrix() function in ArchR to predict the cell types within the scATAC data.

## Fatty acid methyl ester identification and quantitative profile

Fatty acid methyl esters (FAME) were prepared using a modified one-step method of Garces and Mancha [66]. FAME were identified and quantified by Shimadzu GCMS-TQ8050 triple quadrupole mass spectrometer with a CI-MS (Shimadzu, Kyoto, Japan) with a BPX 70 column (25 m × 0.22 mm × 0.25 μm; SGE Inc., Austin, TX). An equal weight FAME mixture (462A; Nu-Chek Prep, Inc.) was used to calculate response factors daily. GC conditions: The program was 30 min in total, the column oven temperature was initially held at 80°C and then ramped at the rate of 15°C/min to 170°C and held for 4 min, then ramped to 240°C at the rate of 7°C/min and held for 10 min. The injection temperature was at 250°C on a splitless injection mode with a 1.5 min solvent cut time. Helium was used as carrier gas in a linear velocity flow control mode, with pressure at 123.5 kPa, total flow at 30.0 mL/min, column flow at 1.31 mL/min, linear velocity at 46.2 cm/s, and purge flow at 5.0 mL/min. Argon was the collision gas and also used to pressurize the solvent-mediated chemical ionization source containing acetonitrile source maintained at 20 kPa. Ion source and interface temperature were set at 240°C.

## Supporting information

**S1 Fig. Quality control metrics for each biological replicate.**
(DOCX)

**S2 Fig. Cell type assignment of clusters.**
(DOCX)

**S3 Fig. Differential accessibility of transcription factor motifs and changes in DHA fatty acid in response to VSG.**
(DOCX)

**S4 Fig. Differential accessibility of genes response to VSG in the ING depot.**
(DOCX)

**S1 File. Supplementary methods.**
(DOCX)

## Acknowledgments

The Cornell University Biotechnology Resource Center (BRC) for genomic sequencing. Chris Donahue for nuclei FACS. Seoyeon Lee from the Soloway lab at Cornell University for helpful discussions. Anne McGavigan from the Cummings lab at Cornell University for use of adipose tissues.

## Author Contributions

**Conceptualization:** Paul D. Soloway.

**Formal analysis:** Blaine Harlan, Hui Gyu Park, J. Thomas Brenna.

**Investigation:** Blaine Harlan, Hui Gyu Park, J. Thomas Brenna.

**Methodology:** Blaine Harlan, Hui Gyu Park, Roman Spektor, J. Thomas Brenna.

**Project administration:** Paul D. Soloway.

**Resources:** Bethany Cummings, Paul D. Soloway.

**Supervision:** Paul D. Soloway.

**Validation:** Hui Gyu Park, J. Thomas Brenna.

**Writing – original draft:** Blaine Harlan, Paul D. Soloway.

**Writing – review & editing:** Blaine Harlan, Roman Spektor, Bethany Cummings, J. Thomas Brenna, Paul D. Soloway.

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
