## [Decision Letter · Decision Letter 0]

13 Oct 2021

PONE-D-21-29146Single-cell Chromatin Accessibility and Lipid Profiling Reveals SCD1-dependent Metabolic Shift in Adipocytes After Bariatric SurgeryPLOS ONE

Dear Dr. Soloway,

Thank you for submitting your manuscript to PLOS ONE. After careful consideration, we feel that it has merit but does not fully meet PLOS ONE’s publication criteria as it currently stands. Therefore, we invite you to submit a revised version of the manuscript that addresses the points raised during the review process. Please address the questions raised from both reviewers point-by-point. Please submit your revised manuscript by 11/22/2022. If you will need more time than this to complete your revisions, please reply to this message or contact the journal office at plosone@plos.org. Please include the following items when submitting your revised manuscript:A rebuttal letter that responds to each point raised by the academic editor and reviewer(s). You should upload this letter as a separate file labeled 'Response to Reviewers'.A marked-up copy of your manuscript that highlights changes made to the original version. You should upload this as a separate file labeled 'Revised Manuscript with Track Changes'.An unmarked version of your revised paper without tracked changes. You should upload this as a separate file labeled 'Manuscript'.

We look forward to receiving your revised manuscript.

Kind regards,

Kai Sun, MD, PhD

Academic Editor

PLOS ONE

Journal Requirements:

3. As part of your revision, please complete and submit a copy of the Full ARRIVE 2.0 Guidelines checklist, a document that aims to improve experimental reporting and reproducibility of animal studies for purposes of post-publication data analysis and reproducibility: https://arriveguidelines.org/sites/arrive/files/Author%20Checklist%20-%20Full.pdf (PDF). Please include your completed checklist as a Supporting Information file. Note that if your paper is accepted for publication, this checklist will be published as part of your article.

8. We note that Figure 1 in your submission contain copyrighted images. All PLOS content is published under the Creative Commons Attribution License (CC BY 4.0), which means that the manuscript, images, and Supporting Information files will be freely available online, and any third party is permitted to access, download, copy, distribute, and use these materials in any way, even commercially, with proper attribution. For more information, see our copyright guidelines: http://journals.plos.org/plosone/s/licenses-and-copyright.

Reviewers' comments:

Reviewer's Responses to Questions

**Comments to the Author**

1. Is the manuscript technically sound, and do the data support the conclusions?

Reviewer #1: Yes

Reviewer #2: Yes

2. Has the statistical analysis been performed appropriately and rigorously? 

Reviewer #1: Yes

Reviewer #2: Yes

3. Have the authors made all data underlying the findings in their manuscript fully available?

Reviewer #1: Yes

Reviewer #2: Yes

4. Is the manuscript presented in an intelligible fashion and written in standard English?

Reviewer #1: Yes

Reviewer #2: Yes

5. Review Comments to the Author

***Reviewer #1:*** In the manuscript entitled “Single-cell Chromatin Accessibility and Lipid Profiling Reveals SCD1-dependent Metabolic Shift in Adipocytes After Bariatric Surgery ”, Harlan and colleagues utilized single nucleus ATAC sequencing (snATAC) to study the mechanism of energy homeostasis and utilization in adipose tissue after bariatric surgery in mice. The authors showed the functional heterogeneity in the adipose tissues by unsupervised clustering analysis. They claimed that accessibility at some transcriptional factor motif (STAT5A, NFIC and NR1C3) changed in inguinal VSG adipocytes, therefore causing metabolism improvement and inflammation reduction. The profiling by snATAC and lipid profiling suggest reduced accumulation of unsaturated fatty acids produced by SCD1 could be the underlying mechanism of bariatric surgery’s beneficial effects.

Overall, this is an interesting work, which could be a reference for other studies. However, in order to serve as reliable reference, some additional validations of the bioinformatics data are needed. In addition, the sample size is relatively small, the authors may need to address the possibility of the genetic variations among the samples.

Some other concerns are listed below:

1. In the assay for the accessibility at transcription factor motifs, the authors combined all the classes of adipocytes. It might prevent the authors from finding out some subtype-specific transcriptional factors to address the heterogeneity of the mature adipocytes.

2. Validations for some of the key computational findings in the study will significantly increase the impact of this study. For one example, a chromatin-immunoprecipitation qPCR may help to validate identified interactions.  For another example, it will be more convincing if the authors validate the SCD1 changes by qPCR or western blotting.

3. Same amounts (80mg) of adipose tissue were used for analysis. However, there are more VSG cells passing the cutoff (Fig S1A). Does it mean more VSG cells were used for experiments and the cell volume of VSG adipose tissue are smaller than those in SHAM groups? Are there more TAG accumulation in the adipocytes of SHAM mice? If so, the transcriptional alternation could be the confounding effect of the TAG accumulation.

4. Some details are missing in the Methods part. Were the samples processed in the same time or in different batches? Was batch correction analysis performed? How were the samples selected for nuclei isolation after the adipose tissue minced? These details need to be provided to help understanding the results.

5.  “To classify cell types within these clusters, we used gene ontology (GO) enrichment for the top 200 differentially accessible genes from each cluster to provide insight into biological processes (fig 1c and fig S2c).” Eventually, Fig S2c lists top 500 DA genes. Please clarify. 

6. “We then confirmed the cell type assignment using known marker genes (Fig 2d, e).” It  should be Fig 1d,e

***Reviewer #2:*** The manuscript by Harlan, et al. reported a single-cell chromatin accessibility analysis in the subcutaneous and epidydimal fat pads post the VSG surgery. To validate their findings, a further lipid profiling was performed. Authors identified fat depot-specific cell composition and chromatin accessibility, and their different response to the VSG surgery. In particular, the change in chromatin accessibility at Scd1 in the subcutaneous fat might explain the beneficial effects of the VSG surgery in mice. Overall, this is a well-designed and well-executed study. I only have a few minor suggestions.

1. The benefits in mice after the VSG mice need to be shown as supporting data, such as body weight, glucose tolerance, etc.

2. A main criticizing point is that authors failed to discuss the recent paper by Backhahl, et al. (Cell Metab 33, 1869-1882) describing distinct human adipocyte subpopulations and their differential sensitivities to insulin. How these subpopulations correlate to the three mature adipocyte subpopulations identified in the current study?

3. In this study, authors focused on the subcutaneous and epidydimal white adipose tissue. What is the change in the brown fat pad after the VSG surgery?

6. PLOS authors have the option to publish the peer review history of their article (what does this mean?). If published, this will include your full peer review and any attached files.

Reviewer #1: No

Reviewer #2: No

---

## [Author Response · Author response to Decision Letter 0]

24 Nov 2021

Below are the editor’s and reviewers’ requests for changes, and our responses to them immediately after. 

Completed

Completed

3. As part of your revision, please complete and submit a copy of the Full ARRIVE 2.0 Guidelines checklist, a document that aims to improve experimental reporting and reproducibility of animal studies for purposes of post-publication data analysis and reproducibility: https://arriveguidelines.org/sites/arrive/files/Author%20Checklist%20-%20Full.pdf (PDF). Please include your completed checklist as a Supporting Information file. Note that if your paper is accepted for publication, this checklist will be published as part of your article.

Completed

Completed

We intend to provide sequencing data via GEO (Gene expression omnibus) upon acceptance for publication. 

This sentence was removed since it was not a core piece of information.

The present study used archived frozen tissues collected from mice described in a prior study described in ref. 15. Because no live mice were used in the present study, we required no further institutional approvals for animal use. 

8. We note that Figure 1 in your submission contain copyrighted images. All PLOS content is published under the Creative Commons Attribution License (CC BY 4.0), which means that the manuscript, images, and Supporting Information files will be freely available online, and any third party is permitted to access, download, copy, distribute, and use these materials in any way, even commercially, with proper attribution. For more information, see our copyright guidelines: http://journals.plos.org/plosone/s/licenses-and-copyright.

The graphical artwork in figure 1a is original work created by the first author (Harlan B.). They do not include material copyrighted by others. 

Reviewer #1: In the manuscript entitled “Single-cell Chromatin Accessibility and Lipid Profiling Reveals SCD1-dependent Metabolic Shift in Adipocytes After Bariatric Surgery ”, Harlan and colleagues utilized single nucleus ATAC sequencing (snATAC) to study the mechanism of energy homeostasis and utilization in adipose tissue after bariatric surgery in mice. The authors showed the functional heterogeneity in the adipose tissues by unsupervised clustering analysis. They claimed that accessibility at some transcriptional factor motif (STAT5A, NFIC and NR1C3) changed in inguinal VSG adipocytes, therefore causing metabolism improvement and inflammation reduction. The profiling by snATAC and lipid profiling suggest reduced accumulation of unsaturated fatty acids produced by SCD1 could be the underlying mechanism of bariatric surgery’s beneficial effects.

Overall, this is an interesting work, which could be a reference for other studies. However, in order to serve as reliable reference, some additional validations of the bioinformatics data are needed. In addition, the sample size is relatively small, the authors may need to address the possibility of the genetic variations among the samples.

We used the inbred mouse line C57BL/6J to minimize genetic variation among the samples and so we believe three biological replicates for each SHAM and VSG across two depots was appropriate to capture most of the variation among individuals. For each biological replicate, we sorted multiple 96 well plates with cells for scATAC-seq analyses. These are formally technical replicates. 

Some other concerns are listed below: 

1. In the assay for the accessibility at transcription factor motifs, the authors combined all the classes of adipocytes. It might prevent the authors from finding out some subtype-specific transcriptional factors to address the heterogeneity of the mature adipocytes.

We expanded our analyses as the reviewer suggested and report in Fig S3 differential motif accessibility among the adipocyte subtypes (AD1, 2, and 4). Please note that when looking for motif differences among the adipocyte subtypes within a treatment group (VSG or SHAM), some cell counts were too low to confidently call peaks for motif enrichment. It is for this reason that clusters representing AD1, 2, and 4 were combined across treatment groups.

2. Validations for some of the key computational findings in the study will significantly increase the impact of this study. For one example, a chromatin-immunoprecipitation qPCR may help to validate identified interactions. For another example, it will be more convincing if the authors validate the SCD1 changes by qPCR or western blotting.

We agree that ChIP could be an orthogonal validation of the ATAC-seq results showing the most dramatic changes in Scd1 in the AD1 and AD2 cells, but not in other cells in adipose tissue. However, there are at least three details to note. First is that for such data to be best compared to ours, they would need to be single cell data to detect differences in protein binding at Scd1 in AD1 and AD2 cells. ChIP-qPCR applied to bulk tissue would have too low a signal to noise ratio relative to our scATAC-seq data to report meaningful differences at Scd1 in AD1 and AD2 cells. Second, the antibody to choose for ChIP is not obvious. Accessibility differences we observed may be due to one of several TFs or histone modifications, and furthermore, their altered binding may be at an intergenic site, with an indirect effect on accessibility at the Scd1 gene. Third and perhaps most importantly is that ChIP or expression measurements do not report the most physiologically relevant consequences for lipid metabolism that result from changes at Scd1 caused by VSG, and those are reported by our lipid profiling experiments. The profiles are exactly those predicted if decreased accessibility at Scd1 after VSG were functionally relevant, and accordingly, those profiles provide the most meaningful orthogonal validation of our findings. Additionally, previous studies have already shown Scd1 expression is decreased in subcutaneous adipose tissue after bariatric surgery (Jahansouz et al., reference #10). 

3. Same amounts (80mg) of adipose tissue were used for analysis. However, there are more VSG cells passing the cutoff (Fig S1A). Does it mean more VSG cells were used for experiments and the cell volume of VSG adipose tissue are smaller than those in SHAM groups? 

It is possible that there are differences between SHAM and VSG treatments that alter cell numbers, their volume, experimental yield or nuclei quality, but our data do not address those possibilities. We applied the same quality control metrics to data from both groups, and excluded no cells that passed these metrics. This invariably produces differences in the amount of data available from independent samples. We consider the numbers of nuclei we sampled to be high enough to support our findings, and are consistent with other studies in the literature.

Are there more TAG accumulation in the adipocytes of SHAM mice? If so, the transcriptional alternation could be the confounding effect of the TAG accumulation.

We did not directly measure TAG accumulation. TAG molecules can contain both saturated and unsaturated fatty acids of various chain lengths. However, there is evidence that MUFAs produced by SCD1 are preferentially used to form TAG. So, it is possible that the higher ratio of MUFA/SFA in SHAM indicates increased TAG accumulation.

4. Some details are missing in the Methods part. Were the samples processed in the same time or in different batches? Was batch correction analysis performed? How were the samples selected for nuclei isolation after the adipose tissue minced? These details need to be provided to help understanding the results.

The details requested have been added to the methods section. In short, the samples were processed in batches of two (one SHAM and one VSG sample). We also attempted to minimize batch effect due to library preparation by using the same machines for all samples (i.e., thermocycler for PCR, nuclei FACS). 

We used iterative latent semantic indexing (LSI) for dimensionality reduction which can minimize batch effects (https://www.archrproject.com/bookdown/iterative-latent-semantic-indexing-lsi.html). After dimensionality reduction, we did not observe significant variation between our biological replicates so additional batch correction, such as Harmony, was not performed. 

5. “To classify cell types within these clusters, we used gene ontology (GO) enrichment for the top 200 differentially accessible genes from each cluster to provide insight into biological processes (fig 1c and fig S2c).” Eventually, Fig S2c lists top 500 DA genes. Please clarify. 

We apologize for the inconsistency. This has been corrected in the manuscript

6. “We then confirmed the cell type assignment using known marker genes (Fig 2d, e).” It should be Fig 1d,e

We appreciate that the reviewer caught this error. It has been corrected in the manuscript.

Reviewer #2: The manuscript by Harlan, et al. reported a single-cell chromatin accessibility analysis in the subcutaneous and epidydimal fat pads post the VSG surgery. To validate their findings, a further lipid profiling was performed. Authors identified fat depot-specific cell composition and chromatin accessibility, and their different response to the VSG surgery. In particular, the change in chromatin accessibility at Scd1 in the subcutaneous fat might explain the beneficial effects of the VSG surgery in mice. Overall, this is a well-designed and well-executed study. I only have a few minor suggestions.

1. The benefits in mice after the VSG mice need to be shown as supporting data, such as body weight, glucose tolerance, etc.

The supporting data you mention was previously published in McGavigan, et al. (DMM Dis Model Mech. 10(3) 235-243; PMID: 28093508). We used tissues from the same mice that were used in that study. Text in the manuscript has been edited to state this more clearly.

2. A main criticizing point is that authors failed to discuss the recent paper by Backhahl, et al. (Cell Metab 33, 1869-1882) describing distinct human adipocyte subpopulations and their differential sensitivities to insulin. How these subpopulations correlate to the three mature adipocyte subpopulations identified in the current study?

At the time of preparing the manuscript, this paper was not yet published. Integration of this spatial transcriptomics dataset with our own dataset has been performed, described in lines 169-176 and the analysis added to supplementary figure 2. In summary, we found most of our adipocyte clusters we defined using our data were predicted to be human adipocyte 1 (the non-insulin responsive cluster) that Bäckdahl described; however, AD1 had a higher fraction of cells predicted to be similar to insulin-responsive human adipocyte 2 cluster in Bäckdahl’s data. This result was interesting and supports our hypothesis that VSG may increase the proportion of insulin-responsive beige-like AD1 cells which contributes to the beneficial metabolic changes.

3. In this study, authors focused on the subcutaneous and epidydimal white adipose tissue. What is the change in the brown fat pad after the VSG surgery?

This is an intriguing question, and certainly worth exploring experimentally. However, we elected to study changes in the white fat as opposed to brown fat after VSG surgery simply because humans have a much higher ratio of white fat to brown fat than do mice. Therefore, we hypothesized that the adipose-related benefits of VSG in white adipose depots in mice would potentially be more relevant to adipose responses to VSG in humans. The role of classical brown adipose in response to VSG and its importance in human metabolism still needs to be evaluated.

---

## [Decision Letter · Decision Letter 1]

10 Dec 2021

Single-cell  chromatin accessibility  and  lipid profiling reveals  SCD1-dependent  metabolic shift  in  adipocytes induced by bariatric surgery

PONE-D-21-29146R1

Dear Dr. Soloway,

We’re pleased to inform you that your manuscript has been judged scientifically suitable for publication and will be formally accepted for publication once it meets all outstanding technical requirements.

Kind regards,

Kai Sun, MD, Ph.D

Associate Professor

Academic Editor

PLOS ONE

Reviewers' comments:

Reviewer's Responses to Questions

**Comments to the Author**

1. If the authors have adequately addressed your comments raised in a previous round of review and you feel that this manuscript is now acceptable for publication, you may indicate that here to bypass the “Comments to the Author” section, enter your conflict of interest statement in the “Confidential to Editor” section, and submit your "Accept" recommendation.

Reviewer #1: All comments have been addressed

Reviewer #2: All comments have been addressed

2. Is the manuscript technically sound, and do the data support the conclusions?

Reviewer #1: Yes

Reviewer #2: Yes

3. Has the statistical analysis been performed appropriately and rigorously? 

Reviewer #1: Yes

Reviewer #2: Yes

4. Have the authors made all data underlying the findings in their manuscript fully available?

Reviewer #1: Yes

Reviewer #2: Yes

5. Is the manuscript presented in an intelligible fashion and written in standard English?

Reviewer #1: Yes

Reviewer #2: Yes

6. Review Comments to the Author

Reviewer #1: In the revised manuscript, the authors addressed the questions and made the changes about the issues raised in the original review. I do think the revised manuscript well meets the publishable standard of the journal.

Reviewer #2: (No Response)

7. PLOS authors have the option to publish the peer review history of their article (what does this mean?). If published, this will include your full peer review and any attached files.

Reviewer #1: No

Reviewer #2: No

---

## [Editor Report · Acceptance letter]

22 Dec 2021

PONE-D-21-29146R1 

Single-cell chromatin accessibility and lipid profiling reveals SCD1-dependent metabolic shift in adipocytes induced by bariatric surgery 

Dear Dr. Soloway:

I'm pleased to inform you that your manuscript has been deemed suitable for publication in PLOS ONE. Congratulations! Your manuscript is now with our production department. 

Kind regards, 

on behalf of

Dr. Kai Sun 

Academic Editor

PLOS ONE